# Controlling Microenvironments with Organs-on-Chips for Osteoarthritis Modelling

**DOI:** 10.3390/cells12040579

**Published:** 2023-02-10

**Authors:** Louis Jun Ye Ong, Xiwei Fan, Antonia Rujia Sun, Lin Mei, Yi-Chin Toh, Indira Prasadam

**Affiliations:** 1School of Mechanical, Medical and Process Engineering, Queensland University of Technology, Brisbane City, QLD 4000, Australia; 2Center for Biomedical Technologies, Queensland University of Technology, Kelvin Grove, QLD 4059, Australia; 3Max Planck Queensland Centre (MPQC) for the Materials Science of Extracellular Matrices, Queensland University of Technology, Brisbane City, QLD 4000, Australia; 4Centre for Microbiome Research, Queensland University of Technology, Brisbane City, QLD 4000, Australia

**Keywords:** osteoarthritis, cell microenvironment, organ-on-chip, disease models

## Abstract

Osteoarthritis (OA) remains a prevalent disease affecting more than 20% of the global population, resulting in morbidity and lower quality of life for patients. The study of OA pathophysiology remains predominantly in animal models due to the complexities of mimicking the physiological environment surrounding the joint tissue. Recent development in microfluidic organ-on-chip (OoC) systems have demonstrated various techniques to mimic and modulate tissue physiological environments. Adaptations of these techniques have demonstrated success in capturing a joint tissue’s tissue physiology for studying the mechanism of OA. Adapting these techniques and strategies can help create human-specific in vitro models that recapitulate the cellular processes involved in OA. This review aims to comprehensively summarise various demonstrations of microfluidic platforms in mimicking joint microenvironments for future platform design iterations.

## 1. Introduction

Osteoarthritis (OA) remains one of the most prevalent chronic musculoskeletal disorders affecting the mobility of the synovial joints, with an estimated global prevalence of 23% among middle- to old-aged individuals [1]. The disease, while common among the elderly, is also observed in young adults, with a relatively high prevalence of 16% [1,2,3]. As a whole joint disease, OA affects multiple tissues in the joint, including subchondral bone, synovium, menisci, tendons, and ligaments, with cartilage degeneration serving as the hallmark of the disease [4]. The highly complex and heterogeneous nature of OA aetiology between patients poses enormous challenges for early diagnostic modalities and preventative drugs. The OA pathophysiology complexities further contribute to the lack of validated models that systematically study the mechanism and provide a suitable screening platform for drug discovery [5].

Various OA models, including small and large animal models, ex vivo tissue explants, in vitro two-dimensional (2D) models, and 3D cell culture models, have been developed to facilitate the mechanistic understanding of the disease and identify suitable treatments [6]. Despite animal models being the gold standard for modelling OA [7], the interspecies differences between animal models and humans become the limiting factor in extrapolating observed responses to human responses when translating in vivo models to human clinical trials [8,9]. Furthermore, the extensive cost of maintaining and conducting animal studies and ethical concerns have restricted the development of reliable in vivo models that can accurately represent different stages and phenotypes of OA. In vitro and ex vivo models, on the other hand, provide cost- and time-efficient approaches for studying the pathophysiology of OA, including the disease progression mechanism and treatment response at both tissue and cellular levels [7]. What is important to note here is that, to this day, in vitro tissue culture platforms still need improving if they are to provide an accurate response. To date, solutions to enhance the relevance of in vitro tissue cultures for OA include using primary cells for patient-relevant tissue response and controlling the tissue culture environment to retain cell functional phenotypes. Primary cell cultures often lose their tissue phenotype during long-term culture due to the lack of a suitable microenvironment [10]. The limitations of standardised tissue culture platforms in mimicking the joint tissue environment, such as tissue structure, tissue environment, and inter-tissue communications, can reduce the predictive capabilities for OA disease models and drug screening. As such, there remains a critical technological gap for a more physiologically relevant platform to accurately model joint tissue and provide a reliable prediction for drug screening, diagnostics, and disease modelling.

Microscale tissue culture devices have evolved with the development of microfabrication technologies, which enables tissue culture engineers to match physiological tissue environments [11]. These tissue microenvironment factors, including the tissue structure [12], mechanical stresses [13], biochemical environment [14], physicochemical environment [15], and cell–cell communications [12], are known to affect joint tissue phenotype. This class of tissue culture devices is commonly known as microfluidic Organ-on-Chip (OoC) [16,17]. Compared to standard tissue culture well plates, OoCs have been demonstrated to substantially address the limitations of traditional tissue culture platforms [18,19,20,21] in the understanding of joint disease and the development of personalised medicine [22]. Demonstrations of chondrocyte culture in OoC for OA were recently reported [22,23]. However, the complex mechanism of OA requires OoCs to mimic multiple aspects of the physiological microenvironments to ensure patient responses in vitro are predictive [24]. Earlier reviews by Paggi and Banh highlighted the importance of modelling cell–cell communications using OoCs and suggested implementing multi-tissue cultures OoCs [22,25]. However, for OA disease progression, the tissue microenvironment, including tissue structure, extracellular matrix (ECM) content, and physical forces, is essential to maintain the cell phenotype [24]. This technical review aims to provide an overview of currently available engineering techniques to mimic the OA joint microenvironment for studying OA pathophysiology. Section 1 presents an analysis of the various physiological microenvironments provided by current OA models and cross-examines current in vitro approaches in 2D and 3D tissue cultures before examining key strategies demonstrated, to date, on mimicking tissue microenvironments in OoCs. The five critical microenvironmental aspects of the joint tissue, namely the tissue culture configuration, mechanical loading and stresses, biochemical contents, oxygen tension, and cell–cell communication are discussed in this review, followed by the current challenges in OoC in microenvironment mimicry and their potential application in the context of OA.

## 2. Animal Models in OA Disease Modelling and Drug Screening

OA is now considered a systemic disease involving the suffering of the whole joint, including cartilage, subchondral bone plate, and synovium [26]. Animal models have always been considered the gold standard for disease mimicking and drug screening for OA for their inherent similarity with humans in terms of mimicking tissue environments and being useful in studying various aspects of disease progression [27,28]. Numerous animal models for OA have been developed over the past 50 years with respect to the heterogeneity of profiles in human OA [28]. Previous studies identified that certain strains of mice, guinea pigs, rabbits, dogs, and horses could be used to study OA [29]. Aged mice with STR/ort and C57BL/6 [28,29], and STR/ort mice are a widely recognised OA model whose lesions are pathophysiologically derived from growth and development and behave similarly to human OA. One earlier study reported an increase in the occurrence of OA from 18 weeks of age in such mice and described a higher chance of OA in the knee, elbow, ankle, and temporomandibular joint, particularly in male mice [30]. Rabbits are also a viable model for developing a potential treatment for OA. Previous radiological evidence showed 50% of older rabbits (>six years) develop naturally occurring OA, and more than 70% of rabbits beyond nine years develop naturally occurring OA [31]. Albino Dunkin–Hartley or Hartley guinea pigs are also widely used as models to study OA. The advantages of these strains over large animals are that they are similar in pathophysiology to human OA, and the bone growth and development rate is faster, shortening the experimental cycle and cost [32,33]. Compared with post-traumatic OA, the disadvantages of these OA models are mainly related to the longer aging time and slower OA progression.

Post-traumatic OA is correlated with trauma, the most widely studied subtype of OA. In terms of application, this phenotype of OA is mainly studied using invasive animal models to evaluate potential medicine’s therapeutic effect and prognosis. The advantage of invasive animal models over other OA models is that it progresses rapidly, so the research cycle is shorter, more likely to cause severe lesions, highly reproducible, and less costly. Mice models commonly used to study post-traumatic OA included the DMM and MSX by performing meniscectomy on the knee meniscus. Longo et al. provided a comprehensive review of post-traumatic OA animal models [34]. The classification of animal models of post-traumatic OA includes both surgical induction and chemical induction [29]. In principle, invasive surgery induces OA by destroying joint-related structures to alter or destroy joint force lines, initiate inflammation or instability, or change the weight bearing of the joints. At the same time, chemical agents or proinflammatory reagents, including sodium iodoacetate (MIA), papain, quinolones, and collagenase, can be injected into the joint, which are less invasive and induce OA by destroying joint tissue [35]. Chemically induced models, however, have drawbacks, and their effectiveness is questionable as rapidly progressive cell and tissue destruction can lead to atypical joint changes.

Animal models possess five unique advantages compared with existing common in vitro tissue culture models (Figure 1). First, animal models maintain the original 3D environment that the in vitro environment cannot fully support, including porous structures in cancellous bone [36]. In addition to providing a 3D environment, the stratigraphic arrangement of chondrocytes [37], the elasticity of the cartilage matrix [38], and the motion around joints are represented in animal models. Lastly, multiple tissue types (cartilage, bone, and synovium) can be investigated within each animal. Animal models also endure periodic compression in physiological conditions [39], another critical factor in maintaining cartilage and bone homeostasis. Cell–cell interaction [40] and cell signalling pathways [41] play another indispensable role during pathophysiologic status. Previous studies indicated there is an unbreakable septum between bone and cartilage, making their respiratory state, metabolism, blood supply, chemical exchange, and immune responses entirely different [40]. However, current developments revealed a cross-talk between bone and cartilage. Moreover, the extracellular vesicle [42] bridges bone and cartilage cross-talk, enabling information exchange, macrophage recruitment, inflammation, anti-inflammation balance, and other essential characteristics that help keep the joint’s integrity [43].

Overall, animal models used in OA can simultaneously help decipher cell–cell interaction and tissue-tissue interaction [44], cell and tissue metabolism [45,46], chemical gradient change [47], cell signalling, inflammation and anti-inflammation process [48,49], and mechanical loading [50,51]. The drawbacks of animal models do exist in OA where, for instance, many disease-modifying osteoarthritis drugs (DMOADs) failed in Phase 3 clinical trials [44] partly because of the complexities of different OA patterns. Furthermore, the xenotypic response differences between animals and humans are also responsible for the difference in outcome measurement. Finally, the animal models are treated humanely, thereby limiting the research area and depth [52], demonstrating the urgency for in vitro models to mimic the in vivo microenvironment and provide accurate human-relevant drug responses.

**Figure 1 cells-12-00579-f001:**
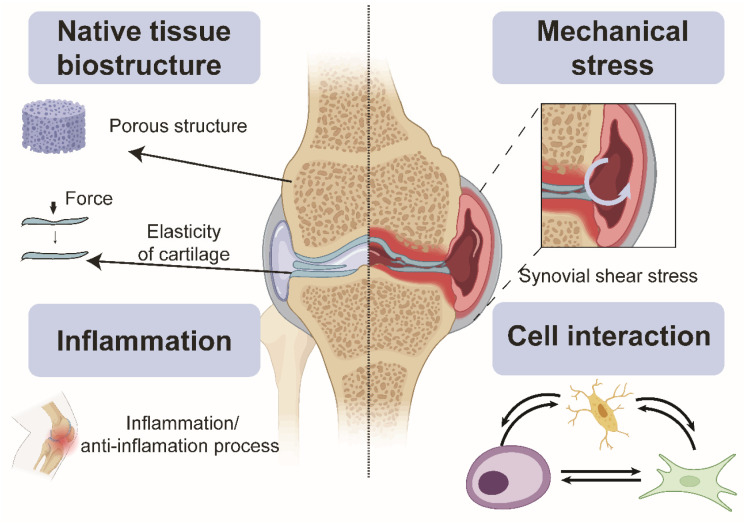
Microenvironmental factors of the joint tissue necessary for modelling joint pathophysiology contributing to OA onset and progression. Created with Biorender.com.

## 3. Conventional In Vitro Models for OA Disease Modelling and Drug Screening

### 3.1. 2D Cultures

In vitro OA models are an excellent substitute for animal models because they can satisfy the criteria to study the progression of OA in vitro [46]. Additionally, in vitro models typically allow for more parallel experimental conditions, which tend to have shorter turnaround times and lower costs [47,48]. While well-plate cultures such as bone and cartilage are highly amenable to high throughput screening platforms [49,50], the 2D monolayer cultures have several changes in the cell phenotype resulting in these platforms being unable to provide human-relevant prediction [51]. Transwell systems make it possible to model cell–cell communication [52]. However, these models are limited to creating co-culture partitioning for up to two tissue types and cannot mimic tissue structure, limiting their use for OA. Compared to an in vivo setting, the lack of mechanosensory stimulation makes interpretation challenging. Chemical gradient and oxygen concentration are also essential for cell culture. Establishing controls of these conditions remains an engineering challenge.

### 3.2. Three-Dimensional Cultures

Three-dimensional culture is more comparable to the OA phenotype than 2D culture as their microenvironment allows more extensive cell-to-cell and cell-to-ECM interactions. For chondrocytes, a previous study showed increased chondrogenic markers (COL2A1, SOX9, and ACAN) in 3D cultures compared to 2D models [36,43]. To mimic the 3D microenvironment, cell pellet culture [44,45], agarose bead [46,47], scaffold [48,49,50], or a mixture of these methods are common in the literature. Pellet culture is the most frequent technique for chondrocytes, in which cells form a densely aggregated 3D cell mass [51]. Scaffolds fabricated using biocompatible materials are popular for their ability to replicate the human body’s native tissue structure.

Beyond mimicking the 3D microenvironment, there is a need to capture the mechanical stress of cartilage and bone during joint movements in the daily activities of OA patients. Simulated joint movement through periodical mechanical loading in tissue constructs is observed to slow the progression of OA. For example, Pötter N. et al. showed physical force exposure on chondrocytes exhibited higher chondrogenic marker expression, such as ACAN, COMP, and COL2, contributing to cartilage matrix maintenance [53].

Biochemical signalling molecules (growth factors and cytokines) also play an essential role in dictating the progression of OA. Inflammatory factors such as IL-6 and IL-1ß [53] are known to trigger cartilage degeneration through the MMP-9 and MMP-13 pathways [53]. The same cytokines also activate macrophages within the synovium, inducing aggrecanase upregulation to further disturb the cartilage matrix [54]. In addition to cytokines, evidence shows the importance of controlling the oxygen microenvironment for cartilage tissue. Specifically, cartilage tissues are found to be physiologically exposed to 1% oxygen levels (hypoxic) [55]. Changes in oxygen content are found to be a key regulating factor for chondrocyte function, especially type II collagen synthesis [56,57]. For example, C Martinez-Armenta et al. demonstrated hypoxic cultures of chondrocytes mimicking physiological oxygen tension and showed a higher chondrogenic phenotype [55]. The control of oxygen further complicates the setup for in vitro culture, especially during co-cultures with other cell types that are physiologically exposed to higher oxygen levels [55]. Co-cultures of chondrocytes and osteoblasts also suggested the role of intercellular communications, which changes the cartilage matrix’s composition during OA [58]. In addition to co-culture with bone, the potential use of mesenchymal stem cells (MSCs) for promoting cartilage formation requires co-culturing chondrocytes [59] to gauge their effectiveness. While these demonstrations showed a possible mechanism of joint tissue physiology of OA, many of the cultures are incubated at ambient oxygen levels to support the MSCs and bone cells, which require higher physiological oxygen levels [55]. Therefore, there is a technological gap for a co-culture platform that allows hypoxic cartilage to be cultured in proximity with other normoxic tissues (bone and MSCs) to capture physiological responses in OA. Recent demonstrations of microfluidic systems mimicking multiple joint tissue physiologies can be an elegant solution to resolve this engineering challenge for studying OA in vitro.

## 4. Organ-on-Chip (OoC) Models for Mimicking Joint Normal and Pathophysiological Microenvironment

Microfluidic organ-on-chip (OoC) is an emerging technology comprising bioengineered human tissues housed within microfluidic devices. OoCs leverage microfabrication technologies to achieve spatio-temporal control over various cell microenvironmental factors at the cellular length scale (between 10–100 microns) (Figure 2) and, therefore, can better mimic the physiological functions of various tissues [60]. The miniaturised and multiplexable nature of OoCs makes them an attractive platform for disease modelling and drug testing, and interest is growing in OoC technologies for generating organotypic tissues, such as the subchondral bone tissues (covered in existing reviews) [22,25]. While the generation of organotypic tissues can improve OoC relevance in OA models and drug screening, control of the tissue environment is essential to maintain native tissue function [61]. Due to the complexity of the joint tissue, as highlighted in Figure 1, many joint OoCs use a deconstructed approach to modelling specific cell environments, allowing the systemic investigation of their role in OA. These strategies can potentially help improve the applicability of in vitro OoC models for disease modelling and drug testing. There has been significant progress in using microfluidics to control various microenvironmental factors relevant to different aspects of joint tissue physiology (Figure 2) in the context of OA, which will be reviewed and discussed in this section, along with bone and cartilage OoCs as most OA OoC models are built on these two models.

### 4.1. Bioengineering 3D Bone and Cartilage Tissues in OoCs

With growing evidence of the importance of 3D cultures in providing physiologically relevant tissue response [66], most of the culture for both bone and cartilage uses various strategies to achieve 3D cultures in microfluidic OoC devices. Techniques modelling 3D tissue structure in OoC are relatively well developed (Figure 3), with the earliest report of OoC design of an OA model noted in 2006 [62]. Many reported 3D cartilage and bone OoC leverages 3D tissue culture techniques developed for other organs. For example, Rosser’s work demonstrated the significance of 3D microfluidic culture where chondrocytes exhibited a higher differentiated cartilage phenotype and osteoarthritis mimicking response to inflammation and treatments compared to 2D monolayers [14]. Similarly, Choudhary et al. also showed that 3D microfluidic ex vivo culture of primary osteocytes could help retain mature osteocytic phenotypes (SOST, FGF23) better than 2D monolayers [67], which plays a crucial role in OA disease progression [68].

For cartilage models, hydrogels were predominantly in OoCs [13,14,69,70,71,72,73,74]. This approach is common in device cultures for the hydrogel to be capable of mimicking the ECM of cartilage. Since cartilage ECM typically contains a mixture of type I and type II collagen, a shift in the collagen content within the ECM could change the function of the chondrocyte [61]. Incorporating cell-laden hydrogels into microfluidic devices can be achieved by either loading liquid-phase hydrogels into a device before setting or by placing crosslinked hydrogels into dedicated chambers.

Liquid-phase hydrogels are typically loaded through a straight channel with microstructures to guide the hydrogel flow. For example, Paggi et al. reported a microfluidic device with channels consisting of lined micropillar arrays, partitioning chondrocytes laden agarose within the microchannel, which were then perfused with culture media [13]. This partitioning allowed 3D cultures of chondrocytes on one side of the microchannels while allowing perfusion on the adjacent chambers [13,14]. This technique relied on the micropillar arrays at specific spatial intervals (typically around 20–30 µm) to hold hydrogels in the liquid phase via surface tension during the loading process [64]. Therefore, this technique would require high-resolution microfabrication foundries. Changing channel geometry is a recent and alternative strategy exploiting hydrogel’s surface tension to enable hydrogel pinning within an open channel. For example, Hou et al. reported a PDMS microchannel device with a 100 µm high device channel positioned next to a microchannel with a higher ceiling [75]. The change in microchannel ceiling functions similarly to the micropillars used by Roger Kamm’s team but alleviates users from the need for high-resolution fabrication techniques [64]. As surface tension depends on the hydrophobicity of the device’s surface material, it is essential to consider the fabrication materials used. With 3D printing adopted increasingly in the fabrication of plastic-based microfluidic devices, the selection of plastic materials and surface modification techniques to modify surface hydrophobicity should be considered during the device design and fabrication process. A similar strategy is also observed for bone cultures [76,77], where, for example, Ma et al. used cell-laden Matrigel to compartmentalise osteocytes in microfluidic devices [76].

A different technique for capturing chondrocyte 3D tissue configuration is densely packing the cells into micro-compartments. Dense-packed chondrocyte pellets, commonly thought to mimic mesenchymal condensation [76], showed an enhanced COL2A: COL1A ratio, which can determine the disease progression in OA. Yang et al. reported the generation of a microfluidic platform with arrays of concave microwells to allow seeded chondrocytes to sediment into the microwells and form 3D micro pellets [78]. The microwell array provides a lower technical barrier for the user to consistently create 3D cell cultures at the microscale, compared with 3D cultures relying on hydrogels. Additionally, this strategy is amendable for various microscopy platforms for high throughput screening and data sampling processes. However, additional design modifications for the microwell-based devices are needed for the platform to be amendable to perfusion cultures. Given that bone tissue does not form dense packing, no work on compact bone cell cultures has been identified for OA models.

Incorporating scaffolds to create 3D tissue models for chondrocytes and bone are also a common strategy [79,80,81]. The stiffness of the scaffold is tuneable to mimic cartilage tissue stiffness, which is essential for maintaining and supporting the differentiation of functional chondrocytes [82]. However, the scaffold material and fabrication process are typically incompatible with standard soft lithography, as creating porous PDMS structures remains challenging. As such, design for scaffold-based microfluidic cultures would consider an additional step to include assembly of the scaffold in the device. The most straightforward method is to fabricate the device and scaffold independently, followed by sequential assembly of the scaffold by plasma bonding, clamping, or chemical sealants. Mekhileri et al. reported using 3D bioprinting to first print PEGT/PBT thermoplastics as a scaffold to house the chondrocytes laden within GelMa layers [83]. With careful device design, these scaffolds can be integrated into microfluidic devices to subject the cultures to fluid shear stress. In another study, Sieber et al. incorporated ceramic-based scaffolds into PDMS devices to develop a microfluidic bone model [79]. One potential advantage of this fabrication strategy is that it reduces the need for high-resolution fabrication typically needed to generate porous structures. With hydrogel loading, the microchambers and channels should be designed to prevent blockage of perfusion channels during gel loading (Figure 3).

Since different aspects of tissue microenvironment can be achieved across the common culture modalities [84], demonstration of OoC culture of chondrocyte cultures often uses a combination of hydrogel with either scaffolds or pellet cultures. For example, Rothbauer et al. cultured densely packed chondrocytes embedded in Matrigel to mimic cell–cell and cell–ECM interactions within the cartilage matrix [12].

**Figure 3 cells-12-00579-f003:**
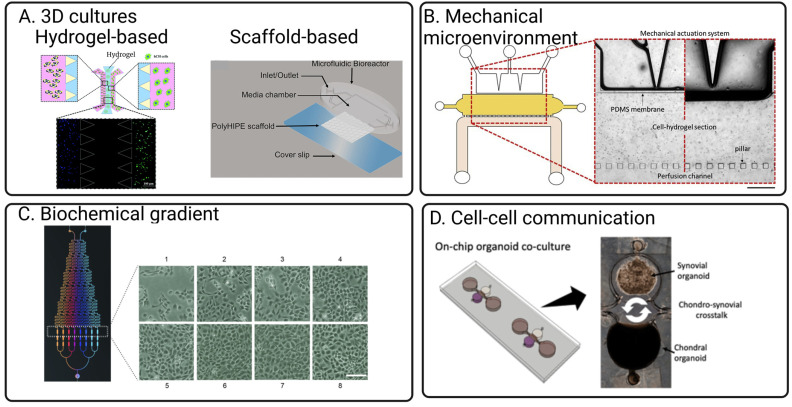
Key strategies implemented in OoC that are adaptable for mimicking joint physiologies for OA models. (**A**) Integrated hydrogels or scaffolds can achieve 3D culture in OoC [73,80]. (**B**) Compressive forces are typically mimicked with an elastic material such as PDMS to allow the channel walls to be compressed using a pneumatic system [13]. (**C**) Through controlling the laminar stream, concentration generators [85] can be integrated on-chip to control concentrations of cytokines and chemokines mimicking OA inflammation responses. (**D**) Cell–cell interactions are typically mimicked through compartmentalised culture models connected with liquid microchannels [12]. Created with BioRender.com. All Figures are obtained under Creative Commons Attribution 3.0 Unported Licence.

### 4.2. Application of Mechanical and Fluid-Induced Shear Stress

With growing evidence of the role of physical activities as a potential treatment of OA, modelling mechanical signalling in microfluidic bone and tissue culture either induced by interstitial fluid motion or through forces subjected during joint loading could provide a suitable animal alternative to screen treatment efficacies for OA. With almost all microfluidic devices holding a volume of less than 500 μL, microfluidic cultures almost always relied on perfusion to facilitate the mass transport of nutrients during cell cultures, which generate fluid-induced shear stress. The shear stress can be controlled by manipulating channel geometries (channel cross-sectional area) and flow rate guided by the Hagen–Poiseuille equation [12], with a lower shear rate generated through channels with a larger cross-sectional area. In cartilage and chondrocytes, abnormal fluid shear stress recapitulates the earmarks of OA, as indicated by the dysregulation of proinflammatory cytokines, matrix metalloproteinases, and pro-apoptotic factors [86,87,88]. As such, the manipulation of shear stress for cartilage and bone OoC to mimic interstitial fluid flow was noted as early as 2006 for OA modelling. In the articular joint, a low fluid shear rate at 2–10 dynes cm^−2^ [89], corresponding to interstitial flow, contributed to the chondroprotective effect on the cartilage tissue and promoted the ERK pathway, essential for osteoblast survival and proliferation [87]. The perfusion of fluids exposes cells to shear stress that could be beneficial for chondrocyte regeneration [90], inflammation response [88], ECM synthesis [88] as well as bone cell morphogenesis [91]. Bao et al. subjected MSC-derived cartilage cells cultured in microfluidic devices with shear stress akin to interstitial fluid flow within the cartilage space (0.05 dynes/cm^2^) [69]. It was noted that cartilage cultures exposed to low shear stress showed improved collagen II and aggrecan synthesis compared to static and 2D cultures.

Apart from liquid shear stress, cartilage and bone tissues are also subjected to consistent mechanical stress during joint motion. Various attempts have been reported to generate mechanical stress within microfluidic devices suitable for cartilage tissue models. For example, Paggi et al. used a pneumatic channel network alongside the cartilage chambers [13]. The chondrocyte can be subjected to physiological periodical mechanical compression of the channel walls against the cartilage cultures mimicking mechanical stress during joint motion by controlling the air pressure in and out of the pneumatic network. The application of rhythmic compression stress is potentially advantageous as a model for aging and metabolic-centric OA pathophysiology. However, it is worth noting that strategies implementing mechanical stress remain reliant on the flexibility of PDMS, which limits the mode of device fabrication. Although the use of plastic and additive manufacturing is gaining momentum as an alternative for microfluidic devices, the development of biocompatible flexible plastics remains slow. One possible way to circumvent this issue is to rely on multi-material device assembly strategies (akin to scaffold devices) to integrate flexible PDMS.

### 4.3. Mimicking Biochemical Environment/Chemical Gradient

One of the hallmarks of OA disease pathology is the inflammation response to the cartilage and bone tissue in both acute and post-traumatic injury. During acute joint damage, the regulation of cytokines, such as IL-1, IL-6, and IL-10, within the joint synovium affects the regulation of cartilage metabolism [92]. In the context of cartilage, tuning both the growth factors and cytokines is crucial to the differentiation process, wound healing [93], and the inflammation response of post-traumatic OA cases. Apart from cytokines, extracellular vesicles (EVs) within the synovial joint are known to regulate cartilage and joint tissue functions [42]. Emerging evidence has indicated that synovium and or articular cartilage-secreted EVs carrying a wide variety of biomolecules, including mRNA, protein, microRNA, and inflammatory cytokines, are found to facilitate tissue cross-talk during OA [43]. For instance, Wu et al. demonstrated that the subchondral osteoblast from sclerotic regions of OA patients secreted EVs containing miR-210-5p that is responsible for manipulating chondrocyte phenotype by altering the expression of ECM production and catabolic activities in vitro, and this miR expression was significantly enhanced in OA patients [94]. Another study revealed that upregulation of long non-coding RNA plasmacytoma variant translocation 1 (PVT1) in isolated EVs from whole blood of OA patients induced apoptosis, inflammation responses and collagen degradation of chondrocytes by modulating the Toll-like receptor 4/NF-κB pathway through an miR-93-5p/HMGB1 axis [95]. It is, therefore, highly desirable to develop a physiologically similar culture model that can potentially be used to examine the transport and uptake of the secreted EVs. The growing interest in utilising EVs for OA therapy would require a culture platform that can mimic a patient’s native biochemical environment as well.

Liquid mixing within microfluidic devices remains diffusion dominant due to the inherently low Re number (<100). It is common for microfluidic channels to contain flow dividers separating fluid streams to generate a chemical gradient, subjecting cells to a range of chemical concentrations depending on their position. Several microfluidic devices use this mechanism to generate polarised tissue cultures [14] and perform dose-response studies for drugs [85]. These devices typically consisted of two separate inlets (one for the drug/chemical of choice and the other as diluent), linked by branches downstream of the inlet, generating a range of chemical concentrations [96]. Uniform mixing at the end of each flow outlet can be achieved by creating micro architectures or changing channel geometries to induce streamlined mixing [85]. A controlled, non-uniform distribution of cytokines and growth could help for zonal differentiation of both chondrocyte and osteocyte phenotype—vital for capturing different aspects of a joint [97] within one device.

To date, several demonstrations have integrated these strategies to tune growth factor exposure to the cartilage culture for OA models, which was covered extensively in the previous review by Banh et al. [25]. For example, Tian et al. incorporated a concentration generator upstream of chondrocyte cultures to investigate interleukin-mediated chondrocyte proliferation parallelly. Within one setup, Tian et al. demonstrated the capability to screen multiple dosages of interleukin to induce cartilage repair [96]. However, the footprint of these microfluidic devices will be relatively large due to the need to incorporate branched fluid channels and mixing mechanisms. In addition to a larger OoC footprint, for finer control of concentration, finer placement of flow dividers at 10 µm resolution might be needed [96], which limits the device design to microfabrication. An alternative is to leverage hydrogels as a diffusion barrier to control chemokine distribution on-chip [98], achievable by loading hydrogels into a microfluidic channel to form a diffusive barrier. The patterning of the hydrogel can be achieved by manipulating microchannel geometries or micropillar arrays [99] (discussed in an earlier section). The hydrogel diffusion barrier porosity can be tuned by controlling the concentration of the hydrogel [100] and the crosslinking process for controlling diffusion rates of essential biomolecules [101]. This strategy has been highly successful in partitioning culture media content [102], EVs [103], and oxygen content as well [15].

### 4.4. Localised Control of Oxygen Tension

In addition to controlling soluble growth factors, drugs, and cytokines, there is a need to tune the dissolved oxygen environment for OA. The mimicry of the hypoxia region in microfluidic joint models is particularly important since healthy articular chondrocytes rely on hypoxia to retain their metabolism and matrix synthesis [54,55]. Evidence in human cartilage samples showed HIF-1 and HIF-2 regulate chondrogenesis and cartilage catabolism through the metalloproteinases and VEGF pathway [104]. Hypoxia also helps regulate cartilage matrix catabolic enzymes in bone marrow-derived mesenchymal stem cells [105]. The ability to model oxygen tension on-chip will be pivotal in enabling the setup of OA disease models and as a screening platform for potential stem cell therapies. Few approaches to date have been implemented to control the oxygen tension for chondrocyte culture on-chip, as most hypoxia environments were established under hypoxic incubators [67,78]. While this approach effectively mimics the physiological oxygen tension in vivo, it remains challenging to model the tissue cross-talk of OA pathophysiology where co-culture with mildly hypoxic bone tissues is needed [106]. As a result, there is a need for joint OoC to segregate the microenvironment for both cartilage and bone tissue while allowing small molecule diffusion between these tissue culture chambers.

Strategies for generating hypoxia locally for OoC tissue cultures remain limited [55], with the earliest OA-related OoC work reported in 2021 [15]; this is partially due to the high technical barriers to implementing these platforms into tissue culture works, as many rely on highly specialised microfabrication techniques. However, techniques demonstrated with gut-on-chip, which utilise low oxygen media (generated by flowing media through a low oxygen chamber) or chemical scavengers, can be implemented for OA tissue co-culture with low technical entry barriers. For example, Ingber’s team demonstrated a gut-on-chip device with two separate media inlets (one with low oxygen medium using nitrogen and one with normoxic medium), which feeds into two closely separated culture compartments, creating a culture environment with heterogeneous oxygen tension mimicking the gut physiology [106]. The chemical scavenging approach of controlling oxygen tension was also noted when Kim et al. utilised the perfusion of chemical oxygen scavengers to purge the oxygen level for gut-on-chip models [107].

### 4.5. Mimicking Bone Cartilage Cross-Talk

Various OoC co-culture models were established to capture the cross-talk between chondrocytes, osteoblasts, osteoclasts, and macrophages as a potential OA model (Table 1). With careful channel design geometry, it is possible to create a simplistic co-culture chip that can model tissue cross-talk with their physiological tissue structure [108,109,110]. For example, Oliveria et al. demonstrated a co-culture platform for chondrocytes and monocytes to model the inflammation response and recruitment of monocyte during OA progression [73]. Similar work was shown as well with bone–monocyte cultures [111], synoviocyte–bone [76], and synoviocyte–cartilage [12] co-culture models. In general, approaches in microfluidic co-cultures for OA models encompass partitioned microchambers to segregate the individual cell types with a shared media circulating the chambers for cell–cell cross-talk. Multi-chamber OoCs offer a unique solution for pairing different cell cultures, useful to identify the mechanisms of complex diseases such as OA. For example, cross-talk between bone and cartilage, modelling for cartilage response to inflamed bone has been captured in multiple bone cartilage OoC models [102]. Apart from bone and cartilage cross-talk contributing to OA disease progression, pathological changes in synovium have been associated with pain and disease progression in patients [112,113].

Synoviocytes have been found to play a key role in regulating OA inflammation cascade via proinflammatory factors such as nitric oxide and prostaglandin E [116]. Inflammatory molecules, including IL-1β, TNF-α, and IL-6, expressed by activated macrophage-like synoviocytes, contribute to cartilage damage and bone alterations [117]. At the same time, fibroblast-like synoviocytes from the site of joint pain in knee OA patients exhibit a distinct subset that plays a key role in promoting fibrosis, inflammation, and the growth and activity of neurons [118]. These findings highlighted possible cross-talk between synovium and other tissues in the joint during OA.

Therefore, the pairing of synoviocytes and chondrocytes in OoCs would enable mechanistic investigations into inflammation regulation within the synovium. For example, Rothbauer et al. co-cultured synoviocyte and chondrocyte connected by a perfusion channel [13] to study the paracrine signalling effect of synoviocyte on the sensitivity of chondrocyte to proinflammatory molecules.

OoC co-cultures of bone/cartilage with mesenchymal stem cells (MSCs) can be a potential application to identify stem cell-based therapies. MSC’s immunomodulatory role through EVs has been suggested for application as a treatment option for OA [119,120]. At the time of this review, limited multi-cell co-culture with stem cells on-chips for OA had been observed [104]. Since most MSCs and stem cell cultures are characterised by conventional 2D cultures, it is likely that differentiation and culture of stem cells with controlled microenvironments need to be thoroughly characterised before their co-culturing with other cell types on-chip.

Neovascularisation during OA involves multiple cell types (chondrocyte, osteoblast, and endothelial cells), which can potentially be studied in OoC co-cultures. The neovascularisation process is known to be assisted through VEGF signalling by the osteoblast [121,122]. In addition to osteoblasts, endothelial cells play a significant role in OA neovascularisation through the digestion of the cartilage and inducing vascular formation through the VEGF signalling pathway [121]. The development of microfluidic neovascularisation models largely focuses on other tissue types, as covered in existing reviews [123]. Generally, for neovascularisation, co-culture of endothelium and fibroblasts are required. For example, Jeon’s team patterned HUVECS and fibroblasts in concentric cylinders creating vascular networks [124]. Due to OA neovascularisation involving multiple cell types, modelling in OA angiogenesis would require co-culture with more than three cell types.

There is minimal control of the biochemical environment in many of these models due to the circulation of shared culture media within the culture device (Table 1).

## 5. Current Challenges in OoC in Controlling Tissue Microenvironment

The growing interest in translating OoCs for large-scale manufacturing processes highlights the use of PDMS as OoC fabrication material is no longer ideal for the next generation of OoC, compared to their plastic counterparts such as PEGDA and polycarbonates, heralded as materials compatible with high throughput fabrication and rapid prototyping. This shift in paradigm in fabrication material has significant consequences for microfluidic OoC. In particular, the guided flow of hydrogels requires hydrophobic surfaces (from the PDMS) to generate surface tension between the loaded hydrogels and neighbouring channels. In addition to the reliance on hydrophobic PDMS, many OoCs still depend on flexible PDMS to actuate and induce mechanical stress across tissue cultures. Therefore, additional considerations, such as altering device fabrication to incorporate a flexible membrane, should be considered for joint OoCs in an approach similar to that demonstrated by Ingber’s group [125]. Alternatively, Nordin’s group has also shown 3D printing of thin PEGDA membranes, which are flexible for pneumatic actuation [126]. Apart from plastics compatibilities to large-scale manufacturing, plastics OoC can avoid non-specific surface adsorption of drugs commonly faced in PDMS [127].

Fabrication materials aside, the current OoC setup is still limited to 2 cell type co-cultures. A multi-compartment OoC [128] can be useful for capturing bone cartilage and the synovium-mediated cross-talks, which is helpful when examining the role of EVs in OA. One of the greatest challenges in implementing multiple cell type co-culture is the lack of a common medium for the co-culture platforms. Currently, there are a few solutions to bypass this problem, either by carefully selecting cell types such that they use the same culture media or the culture media of the most sensitive cells or by performing a 1:1 culture mixing. The challenge in these approaches is that they are all constrictive in generating multi-cell type joint OoCs for long-term culture. The selection of cell pairings can pair primary cells with sub-optimal cell lines, reducing their utility in drug screenings. Concurrently, co-culturing with media based on the most sensitive cell lines or mixed media may not be enough to maintain the phenotype and function of all co-cultured cells. Overall, this is a problem of controlling the distribution of biomolecules. Flow-based diffusion gradient generators can control the exposure concentration of culture additives. However, the flow stream does not allow cell cross-talk between each stream. Embedding cells in hydrogels and their supportive growth factors could potentially solve this problem. For example, Lin et al. embedded stem cells within Matrigel with parallel chondrogenic and osteogenic media perfusion to induce simultaneous osteo- and chondral tissue differentiation within the OoC [102].

## 6. Outlook and Conclusions

The complexities of OA pathophysiology mean the modelling of joint tissue can be achieved either by creating organo-typic tissue constructs by carefully selecting cell models or by controlling the tissue microenvironment. Both strategies have yielded varying degrees of success for their potential application in OA disease modelling and drug testing applications, along with their challenges for implementation. At this point, it is still difficult to model all the microenvironments in a single OoC. A systematic study of joint OoC tissue behaviour in relation to animal models could help identify the importance of microenvironment aspects in relation to different stages of OA.

With OoC positioned as the next in vitro model, it is essential for tissue engineers to consider fabrication materials and processes that are suitable for large-scale manufacturing. It is equally important for tissue engineers to consider the user’s perspective in manufacturing, handling, and operating these OoCs to improve their compatibility with common lab equipment, such as incubators and microscopes. Approaches to reduce the need to assemble tubing into OoCs are widely popular for their portability [129,130,131]. Apart from improving the useability of OoC in routine biological workflow, 3D printing and biocompatible resins such as PEDGA [132] and polycarbonate [133] are highly valuable for improving the compatibility of customised OoC with laboratory equipment. The high level of OoC compatibility with hydrogels shows the possible integration of functional hydrogels [134] with different tunability to control the mass transport of cytokines, drugs, and EVs across various joint tissues. Finally, the integration of on-chip biosensors to provide real-time readout could be an advantage in identifying key secreted EVs and cytokines across cartilage, bone, and synovium.

The combination of flexible material with hydrogel cultures of joint OoC is potentially useful for future investigation of different stages of OA. For instance, by controlling the PDMS stiffness and pneumatic pressure, different degrees of mechanical stress can be applied to OoC tissue culture to induce injury, mimicking the onset of OA due to mechanical overload [135]. Separately, hydrogel diffusion barriers, controlling for key inflammatory cytokines such as IL-1, IL-6, and IL-10, can potentially be useful in investigating OA under inflammation conditions [92]. The same setup also has the potential to be deployed as a screening platform to identify EVs’ effectiveness as therapeutics and identify synovialcytes migration.

In conclusion, OoCs today provided an effective tool to mimic and control the tissue microenvironment and allow systematic examination of the OA mechanism, effectively independent of animal models. OoCs offer flexibility to mimic multiple tissue environments and can provide physiologically relevant tissue responses compared to standard tissue culture practices. Adapting new fabrication techniques and biomaterials is expected to add value for OoCs in the drug discovery and disease modelling process.

## Figures and Tables

**Figure 2 cells-12-00579-f002:**
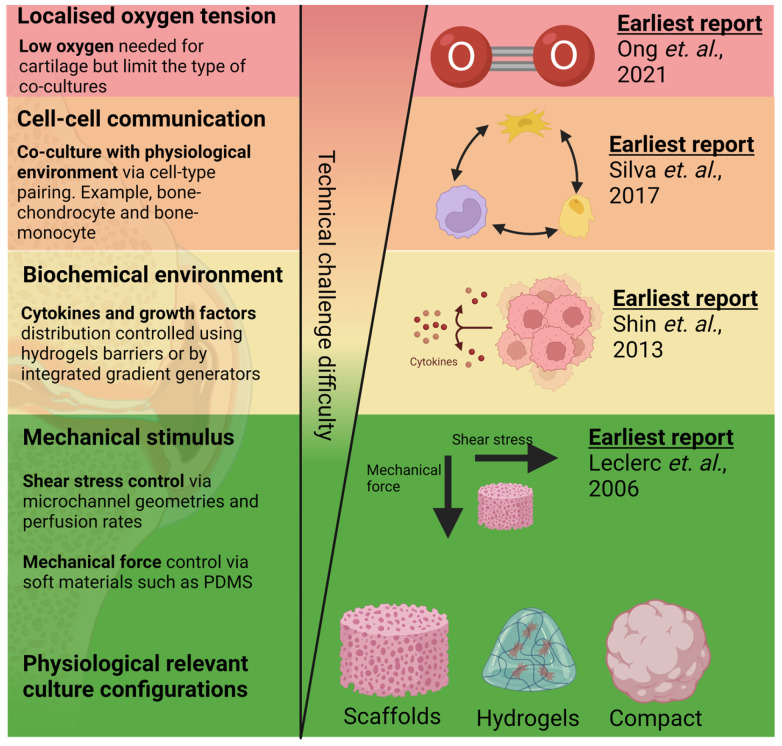
Reported work on mimicking aspects of OA tissue physiology in order of technical challenges in the design, fabrication, and operation of cartilage and bone microfluidic cultures. Techniques to incorporate 3D culture [62] and mechanical [63] and biochemical stimulus [64] are among the earliest techniques developed and have been relatively matured. Alternatively, modelling of oxygen tension control [15] and multi-cell co-cultures [65] are comparatively new techniques and might require greater optimisation for their adoption into routine tissue culture workflow. Created with BioRender.com.

**Table 1 cells-12-00579-t001:** Reported OoC co-culture catering for the modelling of OA.

Co-Cultures	Tissue Culture Mode	Perfusion	Soluble Factor Controls	Ref
Bone|Cartilage	3D hydrogel	Yes	No	[102]
Bone|Cartilage	3D hydrogel	Yes	Yes	[114]
Bone|Cartilage	Scaffold	Yes	Yes	[115]
Bone|Neuron	2D	No	Yes	[65]
Cartilage|Synoviocyte	3D hydrogel	No	Yes	[12]
Bone|Monocytes	3D hydrogel	Yes	No	[98,111]
Bone|Synoviocyte	3D hydrogel	Yes	Yes	[75]

## Data Availability

Not applicable.

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
