# Peer review of "Controlling Microenvironments with Organs-on-Chips for Osteoarthritis Modelling"

_cells, 2023, doi:10.3390/cells12040579_

Round 1

Reviewer 1 Report

Recent development in microfluidic organ-on-chip (OoC) systems have demonstrated various techniques to mimic and modulate different microenvironmental factors relevant to joint pathophysiology physiology. The present review makes an interesting summary of various works of microfluidic platforms in mimicking joint physiology. However, there are still some problems in the current review.

1.     As to mimicking the bone-cartilage crosstalk in Page 9, the authors need to clearly clarify the OoC systems to study the neovascularization and neo-innervation.

2.     Because the effects of extracelluar vesicle (EVs) on the joint pathophysiology have been widely reported recently, the authors need to review the use of OoC systems to study EVs’s effects on cartilage and bone cells.   

3.     Page 2,Compared to standard tissue culture well plates, OoCs demonstrated high potential to overcome the limitations of traditional tissue culture platforms for understanding joint disease and development of personalised medicine that have gained increasing attention in recent yearsThe authors need to describe the shortcomings of standard tissue culture well plates.

4.     Page 3,secondary OA is mainly studied using invasive animal models  The authors need to briefly describe animal models that could mimic secondary OA.

5.     Page 5,various attempts on 3D culture have demonstrated the use of hydrogels to control biochemical signalling factors such as growth factors and cytokines that mimics onset of OA.  More explanations are required for the method about the use of hydrogels to control biochemical signalling factors.

6.     Page 6,majority of the culture for both bone and cartilage employs various strategies to achieve 3D cultures in microfluidic OoC devices.Can the authors make a comprehensive comparison about the following three strategies to find the best.

7.     Page 8, Various attempts have been reported in the generation of mechanical stress that within microfluidic devise suitable for cartilage tissue models”  I would suggest that more discussions be included about the reported attempts.

Author Response

Recent development in microfluidic organ-on-chip (OoC) systems have demonstrated various techniques to mimic and modulate different microenvironmental factors relevant to joint pathophysiology physiology. The present review makes an interesting summary of various works of microfluidic platforms in mimicking joint physiology. However, there are still some problems in the current review.

  1. As to mimicking the bone-cartilage crosstalk on Page 9, the authors need to clearly clarify the OoC systems to study neovascularization and neo-innervation.

We thank the reviewer for suggesting the inclusion of neovascularization for the OoC. We have now included neovascularization in OA and has introduced key OoC techniques applicable for neovascularization for the mentioned section at page 11, line 499 – 508:

“Neovascularisation during OA involves multiple cell types (chondrocyte, osteo-blast and endothelial cells) which be potentially studied in OoC co-cultures. The neo-vascularization process is known to be assisted through VEGF signalling by the osteo-blast [125, 126]. Separately, endothelial cells digestion of the cartilage also induces vascular formation through the VEGF signalling pathway [125]. Development of mi-crofluidic neovascularisation models are largely focus on other tissue types, covered in existing reviews [127]. Generally, for neovascularisation, co-culture of endothelium and fibroblast are required. For example, Jeon’s team patterned HUVECS and fibro-blast in concentric cylinders creating vascular networks [128]. Modelling angiogenesis in would require additional bone and cartilage culture chambers relevant for OA.”

  1. Because the effects of extracellular vesicle (EVs) on the joint pathophysiology have been widely reported recently, the authors need to review the use of OoC systems to study EVs’s effects on cartilage and bone cells.

We thank the reviewer for raising the importance of EVs in this review. The role of EV has been incorporated in page 9, line 378- 397:

“Furthermore, the role of extracellular vesicles (EVs) within the synovial joint regulate cartilage and joint tissue functions [42]. Emerging evidence has indicated that synovium and/or articular cartilage secreted EVs carrying a wide variety of biomolecules including mRNA, protein, microRNA and inflammatory cytokines are found to facil-itate tissue crosstalk during OA [42]. For instant, Wu et al demonstrated that OA sclerotic subchondral bone osteoblast secreted EVs containing miR-210-5p are responsible for manipulating chondrocyte phenotype by altering the expression of extracellular matrix production and catabolic activities in vitro, and this miR expression was sig-nificantly enhanced in OA patients [94]. Another study revealed that upregulation of long non-coding RNA plasmacytoma variant translocation 1 (PVT1) in isolated EVs from whole blood of OA patients induced apoptosis, inflammation responses and collagen degradation of chondrocytes by modulating the Toll-like receptor 4/NF-κB pathway through an miR-93-5p/HMGB1 axis[95]. Thus, it is highly desirable to develop a physiologically similar culture models that can be potentially useful in examining the transport and uptake of the secreted EVs. Furthermore, the growing interest in utilising EVs for OA therapy would require culture platform that can mimic patient’s native biochemical environment as well.”

Demonstration of controlling EV in OoC in the context is also demonstrated on page 10, line 420-423:

“The hydrogel diffusion barrier porosity can be tuned by controlling the concentration of the hydrogel [101]and the crosslinking process for controlling diffusion rates of essential biomolecules [102]. This strategy has shown great success in partitioning cultural media content [103], EVs [104], and oxygen content as well []”

  1. Page 2,“Compared to standard tissue culture well plates, OoCs demonstrated high potential to overcome the limitations of traditional tissue culture platforms for understanding joint disease and development of personalised medicine that have gained increasing attention in recent years” The authors need to describe the shortcomings of standard tissue culture well plates.

We thank the reviewer for raising this issue. The shortcomings of standard tissue cultures have been comprehensively covered in the existing review and are not a key focus on this review. We have included a brief section and referred the readers to appropriate reviews for this part. Page 2 line 68-71 now reads:

“Compared to standard tissue culture well plates, OoCs demonstrated high potential to overcome the limitations of traditional tissue culture platforms [19-22] for understanding joint disease and the development of personalised medicine that has gained increasing attention in recent years [23].”

We also added more specific aspects of tissue cultures that can benefit from OoC’s strategies to control tissue culture microenvironment in page 5, line 197 – 209:

“The control of oxygen further complicates the setup for in vitro culture especially during co-cultures with other cell types that are physiologically exposed to higher oxygen levels [59]. Co-cultures of chondrocytes and osteoblasts mimicking joint tissue interfaces is known to change cartilage matrix’s composition [62]. In addition to co-culture with bone, potential use of mesenchymal stem cells (MSCs) for promoting cartilage formation re-quires co-culturing chondrocytes [63] to gauge their effectiveness. While these demonstra-tions showed a possible mechanism of joint tissue physiology of OA, many of the cultures are incubated at ambient oxygen level to support the MSCs and bone cells which require higher physiological oxygen levels [59]. Therefore, there is a technological gap for a co-culture platform that allows hypoxic cartilage to be cultured in proximity with other normoxic tissues (bone and MSCs) to capture physiological responses in OA. Recent demonstrations of microfluidic systems mimicking multiple joint tissue physiologies can be an elegant solution to resolve this engineering challenge for studying OA in vitro.”

  1. Page 3,“secondary OA is mainly studied using invasive animal models”  The authors need to briefly describe animal models that could mimic secondary OA.

Known animal model is added to page 3 line 116 – 119:

“Known mice models commonly used to study secondary OA included the DMM and MSX by performing meniscectomy on the knee meniscus. Longo et al provided a comprehen-sive review on secondary OA animal models [34].”

  1. Page 5,“various attempts on 3D culture have demonstrated the use of hydrogels to control biochemical signalling factors such as growth factors and cytokines that mimics onset of OA.  More explanations are required for the method about the use of hydrogels to control biochemical signalling factors.

Basic principles of hydrogel loading is presented in page 7, line 265-277:

“This approach relied on the micropillar arrays at specific spatial intervals (typically around 20-30 µm) to hold hydrogels in the liquid phase via surface tension during the loading process [70]. Therefore, this technique would require high-resolution microfabric-cation foundries. Changing channel geometry is a recent alternative strategy exploiting hydrogel's surface tension for enabling hydrogel pinning within an open channel. For example, Hou et al. reported a PDMS microchannel device with a 100µm high device channel positioned next to a microchannel with a higher ceiling [71]. The change in mi-crochannel ceiling functions similarly to the micropillars used by Roger Kamm's team but alleviates users from the need for high-resolution fabrication techniques [70]. Separately, placing set hydrogels into dedicated microchambers also reduces the dependency on mi-crofabrication. However, manually transferring set hydrogels would require further de-sign modification to facilitate this procedure in an aseptic environment. As surface ten-sion depends on the hydrophobicity of the device's surface material, it is essential to con-sider the fabrication materials used. “

Hydrogel usage to control diffusion for joint OoC is presented in page 10, line 420 – 423:

“The hydrogel diffusion barrier porosity can be tuned by controlling the concentration of the hydrogel [101]and the crosslinking process for controlling diffusion rates of es-sential biomolecules [102]. This strategy has shown great success in partitioning cul-ture media content [103], EVs [104], and oxygen content as well [].”

  1. Page 6,“majority of the culture for both bone and cartilage employs various strategies to achieve 3D cultures in microfluidic OoC devices.”Can the authors make a comprehensive comparison about the following three strategies to find the best.

The different aspects each modality of 3D cultures in OoC is presented in details.

For 3D hydrogels (page 6, 252- 258): “For cartilage models, hydrogels were predominantly used to facilitate 3D hydrogels [4, 16, 64-69]. This approach is common in device cultures for the ability of the hydrogel to mimic the ECM of cartilage. Since cartilage ECM typically contains a mixture of type I and type II collagen, a shift in the collagen content within the ECM could change the chondro-cyte's function [60]. The incorporation of cell-laden hydrogels into microfluidic devices can be achieved by either loading liquid-phase hydrogels into a device before setting or by placing set hydrogels into dedicated chambers.”

For pellet cultures (page 7, 285 – 287) “Dense-packed chondrocyte pellets, commonly thought to mimic mesenchymal condensa-tion [75], showed an enhanced COL2A: COL1A ratio, which can determine the disease progression in OA.”

It is currently hard to say which 3D modality offers the best as many 3D OoC culture also utilized a combined approach to improve tissue’s phenotype. This information is presented in page 7, 315-320. “Since different aspects of tissue microenvironment can be achieved across the com-mon culture modalities [82], demonstration of OoC culture of chondrocyte cultures often uses a combination of hydrogel with either scaffolds or pellet cultures. For example, Rothbauer cultured densely packed chondrocytes embedded in Matrigel to mimic cell-cell and cell-ECM interaction within the cartilage matrix [83].”

  1. Page 8, “Various attempts have been reported in the generation of mechanical stress that within microfluidic devise suitable for cartilage tissue models”  I would suggest that more discussions be included about the reported attempts.

Principles of controlling shear stresses is presented in page 8, line 339-341: “The shear stress can be controlled by manipulating channel geometries (channel cross-sectional area) and flow rate guided by the Hagen–Poiseuille equation [84], with lower shear rate generated through channels with larger cross-sectional area.”

Examples for controlling shear stress is presented in page 8, 351-353: “Bao et al. subjected MSC-derived cartilage cells cultured in microfluidic devices with shear stress akin to interstitial fluid flow within the cartilage space (0.05 dynes/cm2) via controlled design of channel geometry and perfusion rate [91].”

Reviewer 2 Report

Dear,

This is an interesting article and authors discussed about various demonstrations of microfluidic platforms in mimicking joint physiology and provide a guide in design, material selection and fabrication of microfluidic devices on future platform iterations. There are some comments which be considered before next steps: 

-Introduction is poor and it would be improved

-It would be better to discuss about fate of stem cells in a paragraph. and also, you can discuss about other stem cells in OA such as https://doi.org/10.1155/2022/5304860 (https://www.hindawi.com/journals/sci/2022/5304860/)

- It would be better to add conclusion section

-It would be better to add limitations and future section 

Best Wishes,

-

Author Response

This is an interesting article and authors discussed about various demonstrations of microfluidic platforms in mimicking joint physiology and provide a guide in design, material selection and fabrication of microfluidic devices on future platform iterations. There are some comments which be considered before next steps: 

-Introduction is poor and it would be improved

We thank the reviewer for the feedback. We have restructured the introduction and put more focus on the need of mimicking tissue microenvironment for OA.

Page 2, line 51-59 reads:

“Unfortunately, reported in vitro tissue culture platforms till today still need to be improved to provide an accurate response. To date, solutions to enhance in vitro tissue cultures' rel-evance for OA include using primary cells for patient-relevant tissue response and con-trolling the tissue culture environment to retain cell functional phenotypes. However, primary cell cultures often lose their tissue phenotype during long-term culture due to the lack of a suitable microenvironment. The limitation of standardised tissue culture plat-forms in mimicking the joint tissue environment, such as tissue structure, tissue environ-ment, and inter-tissue communications, can reduce the predictive capabilities for OA dis-ease models and drug screening.”

-It would be better to discuss about fate of stem cells in a paragraph. and also, you can discuss about other stem cells in OA such as https://doi.org/10.1155/2022/5304860 (https://www.hindawi.com/journals/sci/2022/5304860/)

We thank the reviewer for pointing this out and kind suggestion for the reference. The reference is added to the main text. The use of stem cells for OA is added in the introduction. Use of Stemcells for OA is also presented in detail in the OoC tissue cross-talk for OA section in page 11, 492-498.

“OoC co-cultures of bone/cartilage with mesenchymal stem cells (MSCs) can be a potential application to identify stem cell-based therapies. MSCs immunomodulatory role through EVs been suggested for their application as treatment options for OA [123, 124]. At the point of this review, there are limited multi-cell co-culture with stem cells on-chips for OA has been observed [107]. Due to most MSCs and stem cells cul-tures being characterised on conventional 2D cultures, it is likely that differentiation and culture of stems cells with controlled microenvironment needs to be thoroughly characterised before their co-culturing with other cell types on-chip.”

- It would be better to add conclusion section

Conclusion section is now added to the main text at page 13, line 574-580:

“In conclusion, OoCs today provided an effective tool to mimic and control tissue microenvironment allowing systematic examination of OA mechanism effectively independent of animal models. Furthermore, compared to standard tissue culture practices, OoCs offer flexibility to compound multiple environment control to provide physiologically relevant tissue responses. Adapting new fabrication techniques and biomaterials are expected to add value for OoCs in the drug discovery and disease modelling process.”

-It would be better to add limitations and future section 

Current challenges are now included in page 12 line 516-550:

“With the growing interest in translating OoCs for large-scale manufacturing pro-cesses, the use of PDMS as OoC fabrication material is no longer ideal for the next generation of OoC compared to their plastic counterparts such as PEGDA and poly-carbonates, which are heralded as materials compatible with high throughput fabri-cation and rapid prototyping. This shift in paradigm in fabrication material has sig-nificant consequences for microfluidic OoC. In particular, the guided flow of hydro-gels requires hydrophobic surfaces (from the PDMS) to generate surface tension be-tween the loaded hydrogels and neighboring channels. In addition, many OoCs still depend on the flexible PDMS to actuate and induce mechanical stress across tissue cultures. Therefore, additional considerations, such as altering device fabrication to incorporate a flexible membrane, should be considered for joint OoCs in a similar ap-proach demonstrated by Ingber's group [111]. Alternatively, Nordin's group has also shown 3D printing of thin PEGDA membranes which are flexible for pneumatic actu-ation [133].

In addition, the current OoCs setup is still limited to 2 cell-type co-cultures. A multi-compartment OoCs [104] can be useful to capture bone-cartilage and the syno-vium-mediated cross-talks, which can be helpful for examining role of EVs in OA. One of the biggest challenges in implementing multiple cell type co-culture is the lack of a common medium for the co-culture platforms. Currently, there are few solutions to bypass this problem, either by carefully selecting cell types such that they use the same culture media or the culture media of the most sensitive cells or by performing a 1:1 culture mixing. Unfortunately, all these approaches are constrictive toward gener-ating multi-cell type joint OoC for long-term culture. Furthermore, the selection of cell pairings can pair primary cells with sub-optimal cell lines, reducing their utility in drug screenings. At the same time, co-culturing with media based on the most sensi-tive cell lines or mixed media may not be enough to maintain the phenotype and function of all co-cultured cells. All in all, this is a problem of controlling the distribu-tion of biomolecules. Flow-based diffusion gradient generators can control the expo-sure concentration of culture additives. However, the flow stream does not allow cell cross-talks between each stream. Embedding cells in hydrogels and their supportive growth factors could potentially solve this problem. For example, Lin et al. have em-bedded stem cells within matrigel with parallel chondrogenic and osteogenic media perfusion to induce simultaneous osteo- and chondral tissue differentiation within the OoC [103].”

Future section is added before the conclusion in page 12, line: 562-573:

“With OoC being positioned as the next in vitro model, it is also essential for tissue engineers to consider fabrication materials and processes that are amendable for large-scale manufacturing. In addition, it is also important for tissue engineers to con-sider the user’s perspective in manufacturing, handling, and operation of these OoCs to improve their compatibility with common lab equipment (incubators and micro-scopes). Approaches in reducing the need to assemble tubing into OoCs have been widely popular for their portability [134-136]. Furthermore, 3D printing and biocom-patible resins such as PEDGA [137]and polycarbonate [138] for their capability al-lowing customised setup to be amendable to standard pumps. Finally, the high com-patibility of OoC with hydrogels discussed shows the possible integration of func-tional hydrogels [139] with different tunability to control the mass transport of cyto-kines, drugs and EVs across various joint tissues.”

Reviewer 3 Report

The topic of this literature review is on the first view very interesting and important, however, the quality of presentation is rather poor. I found that I did not really learn anything from reading this manuscript. Principles of chips are not well explained and known things like animal models and cell cultures are outlined, but not in a helpful manner. The latter could be reduced to small sequences, inserted into the introduction and the main part dealing with the chips should be better elaborated. The outlook remains a bit superficial. I would wish my a section discussing shortly all the so far remaining limitations of the chips leading to the outlook. Which factors should be included in future to reflect different types and stages of OA. The role of the synovium is not well integrated in the review, only shortly mentioned in table 1.

formatting: check surplus or lacking blanks in the text

abstract:

"to summary", write: "summarize"

introduction

line 31: "affecting the mobility around the synovial joints" write "of" instead of "around"

line 46: "often becomes" rather "become", line 49: "in vivo" cursive like before, the same line 78 and 119, 120, 142

line 50: "different stage" use plural

line 51: "provides" write "provide"

line 69: "for with the complex OA pathophysiology" please check, whether it is correct

line 73: "a multi-tissue cultures OoCs[19]" remove "a"?

line 85-88: add a reference supporting the statement.

do not talk about primary and secondary oA without having it explained...

lines 126-7: "Previous studies think there is an unbreakable septum between bone and cartilage. Thus, their respiratory status, metabolism, blood supply, chemical exchange and immune responses are entirely different." add references, also to lines 130-132.

line 139: "difference" means "differs"

line 150: "to studying" to study or simply studying

line 181 and later: "et al" should be written "et al." throughout the manuscript (for et alii, aliae, Latin)

section 3 (in vitro model) does not provide novel insights

line 233: "Saba also shown" (correct:showed) - this seems not not the correct reference! please check correct citation throughhout the manuscript!

line 236: sentence makes no sense

line 252: "in" is surplus

Fig. 3A too small labeling, not readable

line 333: "reports" singular

line 375: "REF" should be added

there are also a lot of grammar flaws in the rest of the manuscrupt which should be corrected.

Author Response

The topic of this literature review is on the first view very interesting and important, however, the quality of presentation is rather poor. I found that I did not really learn anything from reading this manuscript. Principles of chips are not well explained and known things like animal models and cell cultures are outlined, but not in a helpful manner. The latter could be reduced to small sequences, inserted into the introduction and the main part dealing with the chips should be better elaborated. The outlook remains a bit superficial. I would wish my a section discussing shortly all the so far remaining limitations of the chips leading to the outlook. Which factors should be included in future to reflect different types and stages of OA. The role of the synovium is not well integrated in the review, only shortly mentioned in table 1.

We thank the reviewer for the useful comments to improve the manuscript. We have made the following changes based on the comments to improve the manuscript.

  1. The manuscript introduction is restructured to focus on the microenvironment controls of OoC for OA. Specifically, the need to tissue environmental control to maintain cartilage and bone cell phenotype in page 2, line 51-59.
  2. Principle and demonstration of gel loading in OoCs are now included in page 6, line 264-277.
  3. Principle of shear stress control is added page 8, line 399-341.
  4. Added current challenges and future challenges into the manuscript, see reviewer 2, point 4
  5. The role of synovium in OA is added onto page 11, line 479-486: “Synoviocytes have been found to play a key role in regulating OA inflammation cascade via pro-inflammatory factors (including nitric oxide, prostaglandin E) [119]. Inflammatory molecules including IL-1β, TNF-α and IL-6 expressed by activated macrophage-like synoviocytes contribute to cartilage damage and bone alterations [120]. Furthermore, fibroblast-like synoviocytes from the site of joint pain in knee OA patients exhibit a distinct subset that plays a key role in promoting fibrosis, inflam-mation and the growth and activity of neurons[121].These findings emphasise possi-ble cross-talk between synovium and other tissues in joint during OA.”
  6. Potential microenvironment factor control for future OA OoC models is also discussed in page 13, lin 574-591: “Combination of flexible material with hydrogel cultures of joint OoC is potential-ly useful for future investigation of both primary and secondary OA. Through control-ling the PDMS stiffness and pneumatic pressure, different degrees of mechanical stress can be applied on to OoC tissue culture to induce injury, mimicking onset of primary OA. Separately, hydrogel diffusion barriers, controlling for key inflammatory cytokines such including IL-1, IL-6 and IL-10 can potentially useful to study second-ary OA [92]. The same setup also have potential to be deployed as screening platform to identify EVs’ effectiveness as therapeutics and identify synovialcyte migration.”
  7. Formatting check is perform to identify redundant blanks spaces.
  8. Manuscript is sent to editing services to proofread grammatical errors.

formatting: check surplus or lacking blanks in the text

abstract:

"to summary", write: "summarize" - Corrected

introduction

line 31: "affecting the mobility around the synovial joints" write "of" instead of "around" – “of” changed to “around”

line 46: "often becomes" rather "become", line 49: "in vivo" cursive like before, the same line 78 and 119, 120, 142. "often becomes" changed to "become", all latin words are italicized.

line 50: "different stage" use plural - corrected

line 51: "provides" write "provide" - corrected

line 69: "for with the complex OA pathophysiology" please check, whether it is correct. - Sentence is removed and restructured

line 73: "a multi-tissue cultures OoCs[19]" remove "a"? – a is removed

line 85-88: add a reference supporting the statement. – Reference added

do not talk about primary and secondary oA without having it explained... – The idea of primary and secondary OA is removed and rephrased to avoid confusion.

lines 126-7: "Previous studies think there is an unbreakable septum between bone and cartilage. Thus, their respiratory status, metabolism, blood supply, chemical exchange and immune responses are entirely different." add references, also to lines 130-132. – Reference added.

line 139: "difference" means "differs" - corrected

line 150: "to studying" to study or simply studying - corrected

line 181 and later: "et al" should be written "et al." throughout the manuscript (for et alii, aliae, Latin)  - corrected

section 3 (in vitro model) does not provide novel insights – Limitation of in vitro models added in page 5, 197-209

line 233: "Saba also shown" (correct:showed) - this seems not not the correct reference! please check correct citation throughhout the manuscript! – Citation are all cross referenced

line 236: sentence makes no sense – Corrected to “For cartilage models, hydrogels were predominantly used in OoCs [4, 16, 64-69].”

line 252: "in" is surplus - removed

Fig. 3A too small labeling, not readable – Figure is replaced with better, readable words

line 333: "reports" singular - corrected

line 375: "REF" should be added – referenced added

there are also a lot of grammar flaws in the rest of the manuscrupt which should be corrected - Manuscript is sent to editing services to proofread grammatical errors.

Round 2

Reviewer 2 Report

Dear Editor,

The revised version is acceptable.

Best Regards,

Author Response

We would like to thank the reviewer for the time and effort in making the manuscript impactful. 

We did not note any additional changes and suggestions raised by the reviewer and have proceeded to check the text of the manuscript for further grammatical errors.

Reviewer 3 Report

The authors supplemented substantially the section of the organ on chip so that more valuable information is provided. They addressed my previous comments. Hence, this review makes sense. Nevertheless, there some inconsistencies in grammar etc., examples are listed below. Therefore, the ENGLISH should be improved.

Line 132: „type“ write plural „types“

Next line: „within each animals“ singular?

Line 190: „Specifically it is the inflammatory factors such as IL-6 and IL-1ß [56] which are known to trigger cartilage degeneration through the 191 MMP-9 and MMP-13 pathway“ I think the grammar should also be corrected

The same in line 202:“ Co-cultures of chondrocytes and osteoblasts mimicking joint 202 tissue interfaces is known to change cartilage matrix’s composition“

Line 256: please correct „mimicing“

Line 260: is „set“ surplus?

Line 251: „Choudhary“ he is not the only author add „et al.“, the same in line 317: „Rothbauer“, also line 491

Line 388 „Wu et al“ write „et al.“

Line 380 „is known“ write „are known“

Line 387: „extracellular matrix“ was already abbreviated before, do it consistently (e.g. line 319)

Line 487: add blank

Author Response

Response to reviewer’s comment:

The authors supplemented substantially the section of the organ on chip so that more valuable information is provided. They addressed my previous comments. Hence, this review makes sense. Nevertheless, there some inconsistencies in grammar etc., examples are listed below. Therefore, the ENGLISH should be improved.

Line 132: „type“ write plural „types“

Next line: „within each animals“ singular?

We thank the reviewer for pointing out the error. The mistakes are corrected. Line 132 “type” is corrected to “types” and line 133 “animals” is corrected to “animal”

Line 190: „Specifically it is the inflammatory factors such as IL-6 and IL-1ß [56] which are known to trigger cartilage degeneration through the 191 MMP-9 and MMP-13 pathway“ I think the grammar should also be corrected.

We thank the reviewer for pointing out the grammatic error. The sentence is restructured to improve clarity. Line 190 to 192 now reads “Inflammatory factors such as IL-6 and IL-1ß [56] are known to trigger cartilage degenera-tion through the MMP-9 and MMP-13 pathway [57].”

The same in line 202:“ Co-cultures of chondrocytes and osteoblasts mimicking joint 202 tissue interfaces is known to change cartilage matrix’s composition“

We thank the reviewer for pointing out the grammatic error. The sentence is restructured to improve clarity. Line 202 – 203 now reads “Co-cultures of chondrocytes and osteoblasts also suggested the role of intercellular com-munications which changes the cartilage matrix’s composition during OA [62].”

Line 256: please correct „mimicing“

We thank the reviewer for pointing out the mistake. The typo is corrected in line 256

Line 260: is „set“ surplus?

We thank the reviewer for pointing out misleading term. We have changed the term to “crosslinked” hydrogels in line 260.

Line 251: „Choudhary“ he is not the only author add „et al.“, the same in line 317: „Rothbauer“, also line 491

We thank the reviewer for pointing out this mistake. We have corrected the error to include “et al.” for the cited works.

Line 388 „Wu et al“ write „et al.“

We thank the reviewer for pointing out this mistake. We have corrected the error to include “.” for the cited work.

Line 380 „is known“ write „are known“

Grammar error is corrected in line 380 to be consistent with plural.

Line 387: „extracellular matrix“ was already abbreviated before, do it consistently (e.g. line 319)

We thank the reviewer for the suggestion. The abbreviation is included in line 78, and all subsequent “extracellular matrix” term is updated to “ECM” for consistency.

Line 487: add blank

We thank the reviewer for pointing this out. A blank is added in between the sentence.